# Improving Target Sound Extraction via Disentangled Codec Representations with Privileged Knowledge Distillation

**Dail Kim**[1]**, Joon-Hyuk Chang**[1,2*]
[1]Department of Artificial Intelligence
[2]Department of Electronic Engineering
Hanyang University
Seoul, Republic of Korea
{roar27, jchang}@hanyang.ac.kr

## Abstract

Target sound extraction aims to isolate target sound sources from an input mixture using a target clue to identify the sounds of interest. To address the challenge posed by the wide variety of sounds, recent work has introduced privileged knowledge distillation (PKD), which utilizes privileged information (PI) about the target sound, available only during training. While PKD has shown promise, existing approaches often suffer from overfitting of the teacher model for the overly rich PI and ineffective knowledge transfer to the student model. In this paper, we propose Disentangled Codec Knowledge Distillation (DCKD) to mitigate these issues by regulating the amount and the flow of target sound information within the teacher model. We begin by extracting a compressed representation of the target sound using a neural audio codec to regulate the amount of PI. Disentangled representation learning is then applied to remove class information and extract fine-grained temporal information as PI. Subsequently, an n-hot vector as the class information and the class-independent PI are used to condition the early and later layers of the teacher model, respectively, forming a regulated coarse-to-fine target information flow. The resulting representation is transferred to the student model through feature-level knowledge distillation. Experimental results show that DCKD consistently improves existing methods across model architectures under the multi-target selection condition.

## 1  Introduction

Humans possess a selective hearing ability to focus on a specific sound of interest among overlapping sounds. This auditory perception ability has been extensively emulated across various signal processing studies [1–5]. One major approach to achieve this objective is target sound extraction (TSE), which aims to extract one or more desired sound sources from a mixture of multiple sounds, given target clues that indicate the classes of target sounds [5–10], as illustrated in Figure 1 (a). However, the diversity of sound types and acoustic conditions remains a challenge to achieving robust performance. To address this issue, recent studies have explored the use of additional information of target sound, such as timestamps [11, 12], pitch information [13], and multimodal cues [14, 15], to better guide TSE models. A notable work in this direction is the utilization of privileged knowledge distillation (PKD) [12]. This work utilizes a teacher model trained with a timestamp of the target sound as

---

*Corresponding author.

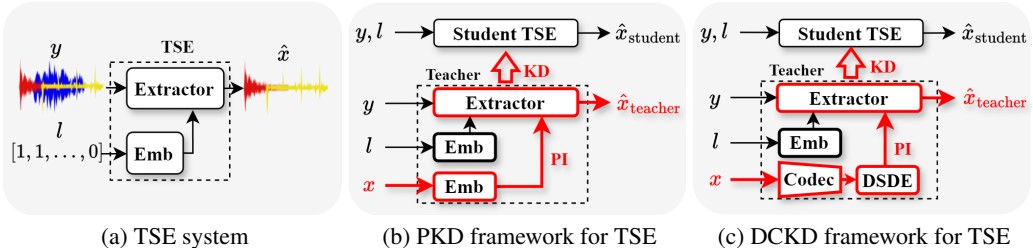

(a) TSE system      (b) PKD framework for TSE      (c) DCKD framework for TSE

Figure 1: A block diagrams of (a) baseline TSE system and the PKD framework of (b) prior network and (c) proposed network for TSE.

privileged information (PI) available only during training and subsequently transfers knowledge to a student model, as shown in Figure 1(b).

While the effectiveness of PKD has been demonstrated in various domains [12, 16–20], there remains substantial room for improvement in how PI is utilized. In particular, highly predictive PI can lead the teacher to overfit to training data, producing high-variance outputs that impair the generalization ability of the student [20]. When applied to TSE, this finding indicates that providing overly rich PI to the teacher may cause performance degradation in the student model.

To mitigate this limitation, we propose *Disentangled Codec Knowledge Distillation* (DCKD), a novel PKD framework for improving TSE. This framework constrains both the amount and the flow of target information by conditioning target class information, provided as a target clue, into the early layers of the teacher model, while class-independent temporal information, used as PI, is conditioned into the later layers (Figure 1(c)). The amount of PI is regulated by leveraging a pretrained state-of-the-art neural audio codec [21] to extract a compressed yet essential representation of the target sound via its encoder and quantization modules. Then, Disentangled representation learning (DRL) is employed to remove class information and extract fine-grained temporal information from the codec representations.

Specifically, the codec representations are passed to a Disentangled Static & Dynamic Encoder (DSDE), which disentangles the representations into static and dynamic factors using designed encoders for each factor. The disentangled representation is assumed primarily based on the static/dynamic factors modeling in [22], but also reflects the global-local information assumption of voice conversion methods [23]. To reinforce disentanglement further, we adopt a mutual information (MI)-based strategy. We minimize MI between the static and dynamic factors using the contrastive log-ratio upper bound estimation method [24], and enhance the representation of the static factor by maximizing its MI with the target sound via a contrastive lower bound estimation method [22, 25]. Meanwhile, the MI between the dynamic factors and the target sound is implicitly maximized through the teacher model training process. Thus, no additional objective is applied to the dynamic factor. The extracted dynamic factors are used to condition the later layers of the teacher model, while the target clue conditions the early layers, forming a coarse-to-fine yet regulated conditioning pipeline of target information. This structure allows the teacher model to first absorb coarse-level class information and then refine its understanding with class-independent temporal information from the PI in later stages, thereby avoiding the injection of overly rich PI all at once. This design mitigates overfitting problem of teacher model and enables more effective feature-level knowledge transfer to the student model. Experimental results on the Kaggle2018-TAU dataset demonstrate that DCKD consistently improves separation performance across various TSE architectures under multi-target selection conditions.

Our key contributions are summarized as follows:

(1) We propose Disentangled Codec Knowledge Distillation (DCKD), a novel privileged knowledge distillation (PKD) framework for target sound extraction that integrates a pretrained neural audio codec with disentangled representation learning (DRL) to facilitate compressed yet essential PI extraction and a coarse-to-fine target information condition scheme.

(2) We address the teacher overfitting issue in PKD by integrating global–local feature disentanglement with DRL, enabling more stable and effective knowledge transfer to the student model.

(3) Experimental results validate that DCKD consistently improves separation performance across various model architectures, achieving notable gains under acoustically challenging conditions, such as mixtures containing 0 to 3 target sounds with varying background noise.

## 2 Related Work

**Target Sound Extraction.** Target sound extraction (TSE) seeks to isolate a target sound source from an audio mixture using user-specified target information provided as a target clue. Early approaches utilized n-hot class labels or reference audio as conditioning inputs [5, 6] to the prior speech separation model, Conv-TasNet architecture [26]. based on these findings, Waveformer [8] was introduced as a real-time streaming model for TSE, demonstrating low-latency performance. More recently, diffusion-based [9] and state-space-based [10] architectures have also been explored. As TSE can be seen as the inherently combined task of sound classification and source separation, subsequent research has explored various forms of target clues to help the model identify target sound, including n-hot class label [5], reference audio enrollment [6, 7], textual queries [27], visual signals [28], and multimodal embeddings [14, 15]. To further enhance TSE performance, several studies have proposed incorporating additional target information, such as timestamps [11, 12] and pitch cues [13], which have shown clear benefits. However, these methods often rely on auxiliary DNNs and may yield suboptimal improvements depending on the quality or type of the additional information. In contrast, our approach enhances performance by effectively transferring PI from the teacher to the student model through principled regulation of PI, without increasing inference complexity.

**Privileged Knowledge Distillation.** Privileged knowledge distillation (PKD) leverages PI available exclusively during training to train a teacher model, which then guides a student model without access to this information [16]. This approach has been widely applied across various domains [17,18,29–32], including audio-related downstream task [17,18,32], due to its ability to utilize privileged information unavailable during inference. However, as noted in [20], providing overly rich PI can cause the teacher model to overfit, leading to ineffective knowledge transfer and degraded student performance. To mitigate this, we employ neural codec to regulate the amount of PI and DRL to enable gradual knowledge transfer. This approach prevents the teacher model from over-relying on privileged information, ensuring effective distillation to the student model.

**Disentangled Representation Learning.** Disentangled representation learning (DRL) seeks to separate latent representations into distinct and interpretable factors, thereby improving model interpretability and generalization [33]. Building on its demonstrated effectiveness across a variety of domains [22, 24, 25], DRL has been applied in speech processing tasks, including voice conversion [23], speech representation learning [34], and target speaker extraction [35]. Recent studies have extended DRL to neural audio codec representations [36], broadening its applicability to general sound processing. In this work, we leverage DRL to improve the effectiveness of PKD for TSE.

## 3 Proposed Approach

### 3.1 Overview

As illustrated in Figure 2, our proposed system consists of two stages:(1) training a teacher model with DRL and (2) training a student model via PKD. Given a mixture signal $y \in \mathbb{R}^L$ of length $L$, an n-hot label vector $l \in [0, 1]^M$ representing the target class(es) among $M$ possible classes, and the target sound $x \in \mathbb{R}^L$, the teacher model is trained using $y$ as input, $l$ as the target clue, and PI derived from $x$, as illustrated in Figure 2(a). To extract PI, target sound $x$ is first passed through a pretrained neural audio codec encoder followed by a quantization module, resulting in a codec representations $c_{1:T}$. This representation is then processed by the proposed DSDE, which separates it into a static factor $s$ and a dynamic factors $z_{1:T}$, following notation of [22]. Among disentangled factors, only the dynamic factors are used to condition the latter part of the teacher model. The static factor thus functions solely to enforce the disentanglement, guiding the dynamic factors to retain class-independent temporal details of the target sound. Conditioning the n-hot vector as a target clue at the front and the dynamic factors at the back of the teacher model enables a regulated, coarse-to-fine flow of target information condition. Once the teacher model is trained, its intermediate feature

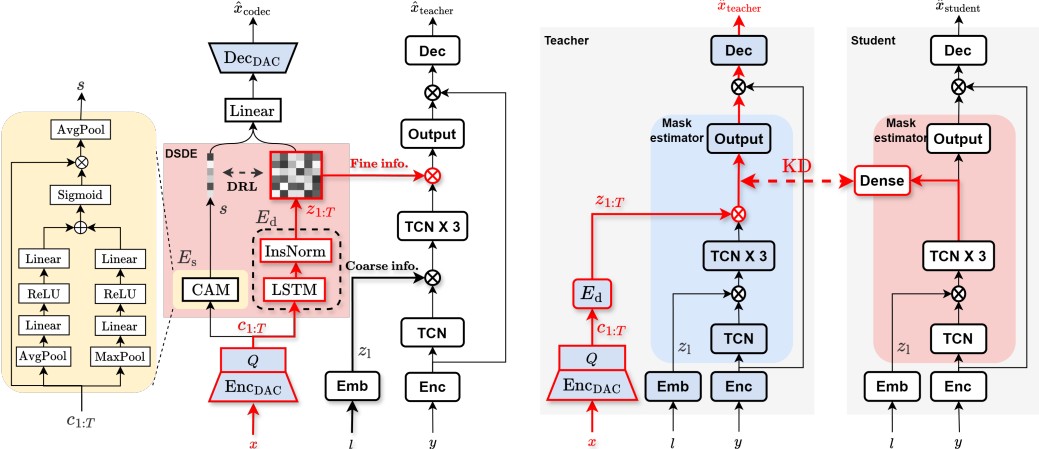

(a) Training teacher model with DRL        (b) Training student model with DCKD

Figure 2: Overview of the proposed DCKD framework, consisting of (a) teacher model training with DRL and (b) student model training with DCKD. During the teacher model training, the model is conditioned on the n-hot vector as a target clue and the disentangled PI extracted from the target sound via a neural audio codec and DSDE. The student model learns from the teacher via feature-level KD, using intermediate latent features. The frozen block is colored in blue.

representations are frozen and used to supervise the student model via a feature-level knowledge distillation loss, as shown in Figure 2(b). This enables the student to learn target information without access to PI during inference, thus preserving efficiency while benefiting from additional information.

## 3.2    Disentangled Codec Representations Learning for Teacher Model

**Disentangled Static & Dynamic Encoder using Neural Codec**    As illustrated in Figure 1(a), the proposed DSDE disentangles neural codec representations of the target sound into a time-invariant static factor and a time-variant dynamic factor. This is achieved via a static encoder $E_s$ and a dynamic encoder $E_d$, each tailored to capture the respective characteristics of the factors. The input to DSDE is the codec representations $c_{1:T} = \{c_1, \ldots, c_T\}$, obtained from a pretrained state-of-the-art neural codec model DAC [21], where $c_i \in \mathbb{R}^{D_c}$ denotes the latent codec representations at frame index $i$ with codec feature dimension $D_c$. The codec representations extraction can be expressed as:

$$c_{1:T} = Q(\text{Enc}_{\text{DAC}}(x)), \tag{1}$$

where $\text{Enc}_{\text{DAC}}$ denotes the encoder of DAC and $Q$ denotes the quantization module.

We assume that the codec representations $c_{1:T}$ can be disentangled into a static factor $s$ and a dynamic factors $z_{1:T}$ under a statistical independence assumption, i.e., $p(s, z_{1:T}) = p(s)p(z_{1:T})$, following prior research [22]. The dynamic factor $z_i$ at each time step $i$ is assumed to be dependent on $z_{<i} = \{z_0, z_1, ..., z_{i-1}\}$ with $z_0 = 0$. Inspired by [23], each frame-wise dynamic factor is assumed to follow a standard Gaussian distribution: $p(z_i) \sim \mathcal{N}(0, I)$ to reduce complexity. Based on these assumptions, the posterior distribution of the disentangled representations to be learned can be factorized as:

$$q(s, z_{1:T}|c_{1:T}) = q(s|c_{1:T})q(z_{1:T}|c_{1:T}) = q(s|c_{1:T})\prod_{i=1}^{T} q(z_i|z_{<i}, c_{<i}). \tag{2}$$

The static factor $s$ is designed to capture time-invariant class identity information, which is modeled using a channel attention module (CAM) [37] to emphasize class-specific information across channels. Note that this static factor is not used as input to the teacher model, instead it is used to enforce the disentanglement of class-related information from the dynamic factor. The dynamic factors $z_{1:T}$ are modeled to capture sequential, local acoustic details using a bi-directional long short-term

memory (Bi-LSTM) network [38]. To remove class information while preserving fine-grained temporal information, we apply instance normalization (IN). IN has been adopted in style transfer and voice conversion to filter out global style attributes [23, 39, 40], such as illumination and contrast in images [39] or pitch and energy in audio [40], while preserving content information. Based on these findings, we hypothesize that IN in our framework can filter out class-specific global features, such as characteristic pitch or energy patterns associated with certain sound classes. Our ablation studies in table 4 confirm that using IN improves performance, supporting our hypothesis that IN facilitates disentanglement of class-related information from the codec representations, thereby enabling more effective application of DCKD. After disentanglement, the static and dynamic factors are concatenated and passed through a linear transformation function $f$, yielding a reconstructed codec representation, which is decoded using the frozen decoder of the DAC $\text{Dec}_{\text{DAC}}$, formulated as:

$$\hat{x}_{\text{c}} = \text{Dec}_{\text{DAC}}(f(s, z_{1:T})). \tag{3}$$

The output of decoder is then optimized to reconstruct target sound using the multi-resolution short-time Fourier transform (STFT) loss [41].

**Mutual Information-Based Disentanglement**   We adopt a MI-based disentanglement approach to encourage the MI between codec representations and each factor while discourage the MI between disentangled factors. However, as noted in [42], disentangled representations can be meaningless without proper inductive biases. To address this, we guide the static factor to retain class-relevant information by minimizing the cosine distance to the embedding vector of the n-hot class label. To minimize the MI between the latent factors $\text{I}(z_{1:T}, s)$, we adopt the variational contrastive log-ratio upper bound (vCLUB) [24] estimator, which is proposed as an upper bound estimate of MI. The vCLUB objective derived based on the modeled factors in (4), is formulated as follows:

$$\text{I}(z_{1:T}, s) \leq \text{I}_{\text{vCLUB}}(z_{1:T}, s) \tag{4}$$
$$= \frac{1}{N} \sum_{i=1}^{N} \left[ \log q(z_{1:T}^{(i)}|s^{(i)}) - \frac{1}{N} \sum_{j=1}^{N} \log q(z_{1:T}^{(j)}|s^{(i)}) \right]$$
$$= \frac{1}{N} \sum_{t=1}^{T} \sum_{i=1}^{N} \left[ \log q(z_t^{(i)}|z_{<t}^{(i)}, s^{(i)}) - \frac{1}{N} \sum_{j=1}^{N} \log q(z_t^{(j)}|z_{<t}^{(j)}, s^{(i)}) \right],$$

where $N$ denotes the number of samples and $q(z_t^{(j)}|z_{<t}^{(j)}, s^{(i)})$ is a variational approximation of the conditional posterior distribution, estimated using a variational approximation network optimized to maximize log-likelihood function. The log-likelihood loss based on the modeled factors in (4) is formulated as,

$$L_{LL} = -\frac{1}{N} \sum_{t=1}^{T} \sum_{i=1}^{N} \log q(z_t^{(i)}|z_{<t}^{(i)}, s^{(i)}). \tag{5}$$

To further improve the representation of disentangled static factor, we maximize its MI with the target sound via MI-lower bound InfoNCE [25], which is contrastive estimation method based on Noise Contrastive Estimation (NCE) [43]. The InfoNCE lower bound for static factor based on augmentation [22] is defined as,

$$\text{I}(z_{1:T}, s) \geq \text{I}_{\text{InfoNCE}}(s, s_{\text{aug}}) = \frac{1}{N} \sum_{i=1}^{N} \left[ \log \left( \frac{e^{\text{sim}(s^{(i)}, s_{\text{aug}}^{(i)})}}{\frac{1}{N} \sum_{j=1}^{N} e^{\text{sim}(s^{(i)}, s_{\text{aug}}^{(j)})}} \right) \right], \tag{6}$$

where $\text{sim}(\cdot, \cdot)$ means cosine similarity function and $s_{\text{aug}}$ is the static factor with content augmentation, generated using the same augmentation strategy as in [22]. We do not apply additional MI maximization to the dynamic factor, as training the teacher model to estimate the target sound implicitly acts as an inductive bias that maximizes MI between the dynamic factors and the target sound.

**Objective Formulation for Teacher Training**   The total objective function for training the teacher model is defined as:

$$L_{\text{Teacher}} = L_{\text{TSE}} + \lambda_{\text{CLUB}}(\text{I}_{\text{vCLUB}} + L_{\text{LL}}) - \lambda_{\text{NCE}}\text{I}_{\text{InfoNCE}} + \lambda_{cos}L_{cos} + L_{\text{STFT}} \tag{7}$$

where $\lambda_{\text{CLUB}}$, $\lambda_{\text{NCE}}$, $\lambda_{\cos}$ are hyperparameters used to balance each objective, and $L_{\text{TSE}}$, $L_{\cos}$, $L_{\text{STFT}}$ represent the TSE loss between $x$ and $\hat{x}_{\text{teacher}}$, the cosine distance between the embedding of the n-hot label $z_l$ and $s$, and the multi-resolution STFT loss between $\hat{x}_{\text{codec}}$ and $x$ [41] for the reconstruction. For the TSE loss, we employ a conditional loss function that applies different objectives based on signal activity to consider the presence or absence of target sounds. Specifically, we compute the negative SDR loss [44] for non-zero (active) targets and a log-scale mean squared error for zero (inactive) targets to penalize hallucinated outputs. The TSE loss for each source is defined as:

$$\mathcal{L}_{\text{TSE}}(x, \hat{x}_{\text{teacher}}) = \begin{cases} 10(\log_{10}(||x||^2) - \log_{10}(||x - \hat{x}_{\text{teacher}}||^2)), & \text{if } ||x||_2 > \varepsilon \\ 10\log_{10}\left(||x - \hat{x}_{\text{teacher}}||^2 + \varepsilon\right), & \text{otherwise} \end{cases} \tag{8}$$

where $\varepsilon$ is a small constant used as a threshold for absence and to prevent numerical instability, which is set to 1e-8.

### 3.3 Disentangled Codec representations Knowledge Distillation for Student Model

After training the teacher model, its privileged knowledge is transferred to the student model through feature-level knowledge distillation. Motivated by [45], we compute the mean squared error (MSE) between the corresponding feature representations of the teacher and student models to transfer the disentangled knowledge:

$$L_{\text{KD}}^{\text{feature}}(z_s, z_t) = ||z_t - g(z_s)||_2^2, \tag{9}$$

where $z_s$ and $z_t$ is the feature representation fron student model and teacher model, and $g(\cdot)$ represents a linear transformation used to align the dimensions for distillation. This feature-based distillation loss is combined with a result-based knowledge distillation loss which can be expressed as,

$$L_{\text{KD}}^{\text{result}} = L_{\text{TSE}}(\hat{x}_{\text{teacher}}, \hat{x}_{\text{student}}), \tag{10}$$

which computes the TSE loss between the outputs of the teacher and student models. To further enhance the effectiveness of knowledge transfer, the student model is initialized with the pretrained weights of the teacher model, followed by fine-tuning. The overall objective function for training student model is formulated as:

$$L_{\text{Student}} = L_{\text{TSE}} + \lambda_{fe}L_{\text{KD}}^{\text{feature}} + \lambda_{re}L_{\text{KD}}^{\text{result}}, \tag{11}$$

where $L_{\text{TSE}}$, $\lambda_{fe}$ and $\lambda_{re}$ is TSE loss between $\hat{x}_{\text{Student}}$ and $x$ are hyper-parameters for the KD objectives.

## 4 Experiments

### 4.1 Experimental Setup

**Evaluation Metric.** We evaluate model performance using signal-to-distortion Ratio (SDR) and Scale-Invariant SDR (SI-SDR) [44] improvements over the mixture signal in decibels (dB) using the BSSeval toolkit [46].

**Dataset.** We constructed a synthetic dataset by combining sound events (SEs) from the Freesound Dataset Kaggle 2018 (FSD Kaggle) [47] and background sounds from the TAU Urban Acoustic Scenes 2019 dataset [48]. The FSD Kaggle dataset provides paired SEs and corresponding class labels across $M = 41$ diverse classes. For dataset partitioning, the train, validation, and eval splits of the FSD dataset were used for training, validation, and testing, respectively. Each mixture was synthesized by randomly selecting $\{3, 4, 5\}$

Table 1: SDR and SI-SDR average of the test set in dB scale.

| | **# of target SE class** | | | |
| | 1 | 2 | 3 | Mean |
|---|---|---|---|---|
| **SDR** | -6.01 | 0.57 | 8.51 | 0.91 |
| **SI-SDR** | -6.10 | 0.54 | 8.50 | 0.87 |

SEs from the FSD dataset, each with a duration of 1.5 to 3 seconds, and adding them at random time positions onto a 5.9-second (260,000 samples) background sound from the TAU dataset. Within each mixture, $\{0, 1, 2, 3\}$ SEs were randomly chosen as targets. The signal-to-noise ratios (SNRs) of the SEs relative to the background were randomly sampled between 15 and 25 dB, with the background level fixed at -30 dB. All audio samples are following original sampling rate, 44.1 kHz. During

training, 2,500 mixtures were generated per epoch through on-the-fly random mixing. For validation and testing, 5,000 fixed mixtures were prepared following the same generation procedure, excluding the zero target sound selection case. Table 1 shows the SDR and SI-SDR of the test set for each number of target SEs.

To validate the generalization capability of DCKD, we further conducted experiments on the ESC-50 dataset [49]. ESC-50 consists of 50 sound classes, each containing 40 clips of 5 seconds. The dataset was split into 20, 10, and 10 clips per class for training, validation, and testing, respectively. Following the same procedure described above, we generated 2,500 on-the-fly mixtures for training and 1,000 fixed mixtures each for validation and testing.

**Implementation.** We employed the Adam optimizer with an initial learning rate of 5e-4. A learning rate scheduler reduced the learning rate by a factor of 0.9 whenever the validation loss failed to improve for three consecutive epochs. The teacher and student models were trained for 500 and 300 epochs, respectively, with early stopping when the models stopped improving for 30 epochs. The model checkpoint with the lowest validation loss was selected for evaluation. During model optimization, the weighting parameter of teacher loss function, $\lambda_{\text{CLUB}}$, $\lambda_{\text{NCE}}$ and $\lambda_{cos}$ were fixed at 1e-5, 1.0, and 10.0 and for student loss function, $\lambda_{fe}$ and $\lambda_{re}$ were set to 1e-4 and 0.6 to balance the values of losses. For the experiments comparing different types of PI, we generated timestamp, pitch, and neural codec representations of the target sound. The timestamp information was derived from the known time indices during mixture generation, using a window size of 64 and a stride of 32, consistent with the encoder of baseline model configuration. Pitch information was extracted using the Parselmouth package [50]. We repeated the extracted pitch values to match the frame dimension of the intermediate features of the teacher model. The same alignment procedure was applied to the neural codec representations. The codec feature dimension was set to 1024, as defined by the pretrained model. All types of privileged information were modeled using a bidirectional LSTM (Bi-LSTM) architecture, identical to that used in the dynamic encoder. All experiments were conducted on a server equipped with four NVIDIA GeForce RTX 4090 GPUs.

**Compared Baseline Configurations.** To validate the effect of the proposed DCKD framework across different DNN architectures, we adopted two distinct models: Sound Selector [5], a widely used baseline for TSE tasks, and Waveformer [8], which was proposed for real-time TSE applications. For the Sound selector, we followed the configuration of [5] composed of 4 repeat of 8 conv1d blocks with learnable encoder-decoder with feature dimension 128, except for the window size, which is 64 adapt to 44.1kHz. The target clue was conditioned after the first repeat and the dynamic factors was incorporated after the final repeat block. We adapted CAM [37] for the static encoder, adding a linear projection layer at the front to match the feature dimensionality and an average pooling layer applied at the output. All pooling operations were implemented as 1D operations along the temporal dimension. To estimate the conditional posterior distribution of the disentangled factors, we jointly train a variational approximation model alongside the teacher model, which consists of four linear layers with ReLU activations for both the mean and log-variance estimators, followed by a final tanh activation to constrain the variance.

For the Waveformer, we utilized the non-causal version introduced in [8]. The stride size for the first 1D convolutional layer was set to 32, and the numbers of encoder and decoder layers were configured as 10 and 1, respectively, following the original implementation. To facilitate stable optimization of PI, we set the encoder and decoder feature dimensions to 256. To match the performance of Waveformer reported in the original paper, we trained it with 10,000 samples per epoch. The target clue was applied following the same approach as in [8], and the dynamic factors was incorporated similarly to the teacher model of the Sound Selector via element-wise multiplication with the output of the mask generator.

## 4.2 Comparison to Privileged Information

Table 2 presents the performance comparison of the proposed DCKD framework under various types of PI using the Sound Selector architecture. The student model trained without any PI showed the lowest performance, with an SDRi of 8.00 dB and SI-SDRi of 7.30 dB. In contrast, teacher models equipped with different forms of PI—such as timestamps, pitch, target sound, and neural codec features—consistently outperformed the student baseline. Notably, using the raw target sound as PI yielded the highest teacher performance, reaching an SDRi of 16.23 dB.

However, when these PIs were transferred to the student model via knowledge distillation, a different tendency was observed. Although all PIs led to performance gains over the baseline student, the effectiveness of distillation did not align with the richness of the PI. In particular, compared to methods using timestamps or pitch as PI, richer forms of PI such as the raw target sound yielded limited improvements for the student model relative to codec-based representations. This shows that overly informative PI can limit the performance gains over the baseline.

For the method employing a compressed target representation obtained via a neural audio codec, although the teacher model performance slightly decreased, the corresponding student model out-performed the one trained with the raw target sound, indicating regulated codec information enables more effective knowledge transfer. Furthermore, although the teacher performance dropped further, our proposed method achieved the highest student performance with 10.40 dB SDRi and 9.42 dB SI-SDRi. These results demonstrate PI regulated through codec and DRL enables more effective distillation to the student model.

Table 2: Comparison to privileged information

| PI Type | Mechanism | SDRi | SI-SDRi |
|---|---|---|---|
| None | Student | 8.00 | 7.30 |
| Timestamp | Teacher | 11.70 | 10.94 |
| | w/ PKD | 9.30 | 8.46 |
| Pitch | Teacher | 11.05 | 10.26 |
| | w/ PKD | 9.61 | 8.71 |
| Target Sound | Teacher | 16.23 | 15.79 |
| | w/ PKD | 9.82 | 8.89 |
| Neural Codec | Teacher | 15.78 | 15.19 |
| | w/ PKD | 10.25 | 9.29 |
| Disentangled Codec | Teacher | 15.32 | 14.77 |
| | w/ PKD | **10.40** | **9.42** |

## 4.3 Comparison to Baselines

Table 3: Comparison of the proposed DCKD framework with different baselines.

| Model | Params (M) | Mechanism | SDRi | SI-SDRi | SI-SDRi | | |
|---|---|---|---|---|---|---|---|
| | | | | | 1-target ($\Delta$) | 2-target ($\Delta$) | 3-target ($\Delta$) |
| **Sound Selector** | 84.44 | Teacher | 15.32 | 14.77 | 19.96 (7.56) | 14.49 (7.23) | 9.61 (5.78) |
| | 6.55 | Student [5] | 8.80 | 7.90 | 12.40 (0.00) | 7.25 (0.00) | 3.83 (0.00) |
| | | w/ DCKD | 10.40 | 9.42 | 14.61 (2.21) | 8.55 (1.30) | 4.84 (1.01) |
| **Wave former** | 81.15 | Teacher | 10.75 | 10.21 | 16.38 (8.46) | 11.16 (6.82) | 8.60 (4.12) |
| | 1.67 | Student [8] | 5.50 | 4.67 | 7.91 (0.00) | 4.34 (0.00) | 1.59 (0.00) |
| | | w/ DCKD | 6.48 | 5.80 | 9.41 (1.50) | 5.31 (0.97) | 2.49 (0.90) |

Table 3 presents a comprehensive comparison between the proposed DCKD framework with teacher-student baselines on two architectures: SoundSelector and Waveformer. Results are reported in terms of SDRi, SI-SDRi, and separation performance under varying target selection scenarios (1-target, 2-target, 3-target). The values in parentheses represent the SI-SDR gain over the student baseline. These results indicate that DCKD consistently improves both architectures architectures regardless of model capacity.

For the Sound selector model, applying DCKD improves the student SDRi from 8.80 dB to 10.40 dB and SI-SDRi from 7.90 dB to 9.42 dB, validating the effectiveness of the proposed framework. In the 1-target condition, DCKD achieves 14.61 dB SI-SDRi, corresponding to a +2.21 dB gain over the student model. Similarly, for 2-target and 3-target cases, DCKD provides gains of +1.30 dB and +1.01 dB, respectively. These consistent improvements indicate that DCKD enhances robustness even as separation difficulty increases.

A similar trend is observed with the lightweight Waveformer model. Although the student baseline performance is lower due to its smaller capacity (1.67M parameters), DCKD improves SDRi from 5.50 dB to 6.48 dB and SI-SDRi from 4.67 dB to 5.80 dB. The improvements were observed across all conditions—1-target (+1.50 dB), 2-target (+0.97 dB), and 3-target (+0.90 dB) selection settings—exhibiting a similar trend to the teacher model. These results demonstrate the robustness of the proposed method, showing that the performance improvements of the student model with DCKD follow the trends observed in the teacher model. In particular, the improvement observed in the compact Waveformer model highlights the potential of DCKD for lightweight models, aligning with one of the primary objectives of knowledge distillation.

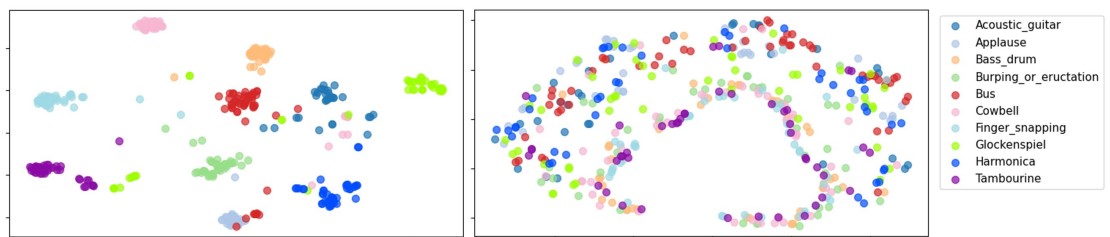

Figure 3: Left: t-SNE visualization of static factor. Right: t-SNE visualization of dynamic factor. The factors are extracted from sound event samples of 10 different sound events from test dataset.

## 4.4 Ablation Study

Table 4 presents an ablation study evaluating the contribution of each component in our proposed DCKD framework, including the neural codec, components for DRL and the feature-level knowledge distillation method.

Table 4: Ablation study of DCKD using the SoundSelector model.

| Configuration | SDRi | SI-SDRi |
|---|---|---|
| w/ DCKD (full) | **10.40** | **9.42** |
| w/o Codec Info | 8.58 | 7.72 |
| w/o $\mathcal{L}_{\text{vCLUB}}$ | 9.88 | 8.94 |
| w/o $\mathcal{L}_{\text{InfoNCE}}$ | 10.34 | 9.37 |
| w/o $\mathcal{L}_{\text{Cos}}$ | 10.11 | 9.15 |
| w/o InstanceNorm | 9.61 | 8.67 |
| w/o MI-based objectives | 9.71 | 8.76 |
| w/o DRL | 10.25 | 9.29 |
| w/o $L_{\text{KD}}^{\text{feature}}$ | 10.38 | 9.43 |

Among all components, removing the neural codec led to the most significant performance drop (-1.82 dB SDRi), highlighting its critical role in providing a compact yet informative representation of the target sound to teacher model. This result confirms the importance of compressing PI via a pretrained codec before distillation.

Disabling the MI based objectives, which include the vCLUB, InfoNCE lower bound, and additional inductive biases such as cosine distance loss and instance normalization, consistently degraded performance, demonstrating that each objective contributes to effective factor disentanglement and inductive guidance. Notably, removing the entire MI-based objectives led to a substantial performance loss (-0.69 dB SDRi), higligthing the benefit of MI-aware DRL. Interestingly, excluding the entire DRL methods, which is the same as the result of the student model with PKD using codec representations as PI, resulted in only a moderate degradation (-0.15 dB SDRi), suggesting that DRL provides more robust benefits when with proper inductive bias.

Lastly, we removed the featur-level KD loss at the feature level to assess its effect. Although the improvement may appear marginal in the results, we observed that featur-level KD loss enables faster convergence when training student model with DCKD by performing meaningful regularisation during the training process.

## 4.5 The Analysis of Disentanglement Learning

To evaluate the effectiveness of our inductive biases for disentanglement, we intuitively visualize the learned static and dynamic factors using t-SNE in Figure 3. This visualization helps assess whether each factor captures the intended properties—static factor capturing target class identity and dynamic factors capturing class-agnostic temporal details. The left panel of Figure 3 illustrates the t-SNE projection of the static factor, exhibiting clear class-wise grouping. This result indicates that the static factor retains sufficient information to discriminate among class identities, validating the effect of the cosine distance loss in guiding class-related representation learning and InfoNCE loss to preserve information of the target sound.

In contrast, the right panel shows the t-SNE distribution of the dynamic factor, which exhibits no evident clustering according to sound class labels. This result is consistent with our objective of removing class identity information from the dynamic factor, preserving only fine grained temporal characteristics.

These results confirm that the proposed DSDE, combined with the MI-based objectives and additional inductive bias, successfully disentangles the codec representations of the target sound representation into static and dynamic factor.

### 4.6 Generalization of DCKD

To evaluate the generalization capability of the proposed DCKD framework, we conducted additional experiments on a ESC-50 dataset, as shown in Table 5. The results demonstrate that applying DCKD consistently improves performance over the baseline student model, validating the robustness and generalizability of the proposed method. The relatively low overall performance and improvement observed in this dataset are likely due to its limited dataset size and the increased number of sound classes (50), which leads to greater inter-class variability that the model must learn.

Table 5: Comparison of student models with and without DCKD on ESC-50 dataset.

| Model | Mechanism | SDRi | SI-SDRi |
|---|---|---|---|
| SoundSelector | Teacher | 10.73 | 10.32 |
| | Student | 4.59 | 3.42 |
| | w/ DCKD | 4.92 | 3.62 |

## 5 Conclusion

In this work, we introduced DCKD, a novel framework that enhances TSE by leveraging disentangled PI. This framework uses target class information and disentangled class-independent temporal information as a condition for early and later layers of the teacher model, forming a coarse-to-fine manner yet regulated information transfer. As the target class information is given as a target clue, our approach mainly focuses on extracting class-independent temporal information from the target sound as a PI for the teacher. PI is extracted by disentangling the neural codec representations of the target sound, guided by an MI-based objective with additional inductive bias. This pre-trained teacher model effectively transfers target-related knowledge to the student model via feature-level knowledge distillation. Experimental results on the Kaggle 2018-TAU dataset demonstrate the effectiveness of our method across diverse scenarios. Furthermore, DCKD mitigates the teacher overfitting problem, which is a key limitation of PKD approaches, suggesting its potential applicability beyond TSE to other domains.

**Limitations and Future work** The proposed method has several limitations. First, the use of a DAC neural codec introduces a considerable computational increase. Although consistent performance improvements were observed, future research exploring lightweight feature compression models and optimized training procedures for the teacher network to enhance efficiency is needed. Second, the current framework assumes access to individual source signals within the mixture during training. This assumption aligns with supervised learning settings but limits applicability to real-world recordings, where isolated target signals may not be available. Therefore, investigating unsupervised or weakly supervised learning strategies to extend DCKD to more practical scenarios can be a key direction for future work.

## Acknowledgments and Disclosure of Funding

This work was partly supported by the National Research Foundation of Korea(NRF) grant funded by the Korea government(MSIT) (RS-2025-00557944), Institute of Information communications Technology Planning Evaluation (IITP) grant funded by the Korea government(MSIT) (No.RS-2025-25443882, Bringing "Her" into the real life: Implementation of the voice AI system enabling sentient conversation) and Institute of Information communications Technology Planning Evaluation (IITP) grant funded by the Korea government(MSIT) (No.RS-2020-II201373, Artificial Intelligence Graduate School Program(Hanyang University)).

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

# A  Implementation Details

**Integration of Course & Fine Information**   The n-hot label is injected into the right after the learnable encoder of the model via element-wise multiplication, where the label vector is first linearly projected to match the channel dimension of the latent feature. The dynamic factors are injected into the latter part through element-wise multiplication, specifically right before the output layer, which consists of a 1D convolution layer.

**Computational Efficiency**   Training time is measured in wall-clock time. The increase in DCKD primarily results from additional forward passes through the teacher model and computation of mutual information-based loss terms.

Table 6: Model size and training time comparison of teacher and student models.

| Configuration | Params (M) | Training Time | GPU Setup |
|---|---|---|---|
| Teacher model | 84.44 | 3 days | $4 \times$ RTX 4090 |
| Student (baseline) | 6.55 | 9 hours | $4 \times$ RTX 4090 |
| Student w/ DCKD | 6.55 | 1 day ($\approx 2.7 \times$ baseline) | $4 \times$ RTX 4090 |

**Implementation Details of Ablation Experiments**   For the ablation study, we used the same teacher and student model architectures when evaluating the effects of feature-level KD loss ($L_{\text{KD}}^{\text{feature}}$) and the cosine distance loss ($\mathcal{L}_{\cos}$).

To validate the effectiveness of the vCLUB method, we excluded $\mathcal{L}_{\text{vCLUB}}$, $\mathcal{L}_{\text{LL}}$ and variational approximation model used for modeling the conditional posterior distribution.

For the MI-based approach ablation, we excluded all MI-based losses: $\mathcal{L}_{\text{vCLUB}}$, $\mathcal{L}_{\text{LL}}$ and $\mathcal{L}_{\text{InfoNCE}}$ to isolate the impact of the MI-based disentanglement strategy.

To assess the contribution of the codec representations, we replaced the codec model with a simple encoder–decoder architecture consisting of a 1D convolutional layer and a transposed 1D convolutional layer. Both the encoder and decoder employed a kernel size of 512, a stride of 256, and a feature dimension of 128.

# B  Ablation Study on Waveformer

Table 7 presents the ablation study results of the Waveformer model, evaluating the effect of each component in the proposed DCKD framework. In most cases, the Waveformer model shows a clear performance degradation below the baseline student model, unlike the SoundSelector-based ablation study, where most configurations still outperformed the baseline. Given that Waveformer has lower model capacity than SoundSelector, this degradation suggests that excessive PI interferes with the convergence of lightweight models. Interestingly, the model without MI-based objectives achieves better performance than the model with DCKD. These results can be explained by the fact that MI estimation increases optimization difficulty and may introduce instability, particularly in compact models such as Waveformer. As a result, using only the Cosine

Table 7: Ablation study of DCKD using the Waveformer model.

| Configuration | SDRi | SI-SDRi |
|---|---|---|
| w/ DCKD (full) | 6.48 | 5.80 |
| w/o Codec Info | 4.04 | 3.11 |
| w/o $\mathcal{L}_{\text{vCLUB}}$ | 4.13 | 3.37 |
| w/o $\mathcal{L}_{\text{InfoNCE}}$ | 5.39 | 4.73 |
| w/o $\mathcal{L}_{\text{Cos}}$ | 3.85 | 3.00 |
| w/o InstanceNorm | 3.92 | 3.21 |
| w/o MI-based objectives | **6.52** | **5.82** |
| w/o DRL | 4.98 | 4.09 |
| w/o $L_{\text{KD}}^{\text{feature}}$ | 6.41 | 5.68 |

similarity loss (Cos) and Instance Normalization (IN) for DRL can lead to more stable training and improved performance in such lightweight architectures. Meanwhile, removing the $L_{KD}^{\text{feature}}$ term

results in only a slight performance drop (6.41 dB / 5.68 dB), following a similar trend to that observed in SoundSelector.

## C    Results on language-query based TSE model

Table 8 shows the results of applying DCKD to the language query–based TSE model, AudioSep [51].

**Implementation**    We used the AudioCap dataset [52] for training, and the test set sourced from [51]. All configurations were kept identical to the original AudioSep setup, except for the sampling rate. While AudioSep originally operated on 32 kHz audio, the sampling rate was set to 44.1 kHz to match the oper-ational range of the DAC, and the model was trained accordingly. For evaluation, the test set was upsampled to 44.1 kHz.

Table 8: Performance comparison of AudioSep models

| Model | Mechanism | SDRi | SI-SDRi |
|-------|-----------|------|---------|
| AudioSep | student | 6.97 | 5.55 |
|  | w/ DCKD | 6.96 | 5.60 |

**Results**    The performance showed a slight improvement in SI-SDRi but remained lower in SDRi. This marginal gain can be attributed to two main factors. First, the caption text often includes not only coarse-level semantic information but also additional temporal cues (e.g., "then," "continuous"), which may have interfered with the intended functioning of DCKD. Second, unlike the baseline models that condition label information only once at the early stage, AudioSep conditions the query information throughout the network using FiLM layers. During teacher model training, fine-grained information is conditioned at the final stage by modulating the feature representations, which cannot be considered as a coarse-to-fine conditioning scheme. Therefore, the limited performance gain is likely due to the structural difference between the conditioning mechanism of the AudioSep and the intended design of DCKD.

## D    Class-Wise Visualization Result

Figure 4 presents the SI-SDR results for all 41 target sound event (SE) classes, comparing the input mixture, the baseline Sound Selector, and the Sound Selector with DCKD applied. Our method consistently outperforms the baseline across most classes, except for "Knock" and "Scissors". In particular, substantial improvements are observed for the classes such as "Tambourine," "Bass drum," and "Saxophone." These results demonstrate the robustness and effectiveness of the proposed framework in enhancing extraction performance across a diverse range of target sound classes.

## E    Sample Spectrograms for Multi-Target Class Selection Scenarios

Figures 5, 6, 7 and 8 present sample spectrogram visualizations. Each figure includes the input mixture, the output of the baseline Sound Selector, the output of the baseline with DCKD applied, the output of the teacher model, and the oracle target signal. These visualizations are provided under varying numbers of target class selections, demonstrating robust improvement over the baseline in both interference sound suppression and target sound detection performance. Example audio files corresponding to the figures are included in the supplementary zip file.

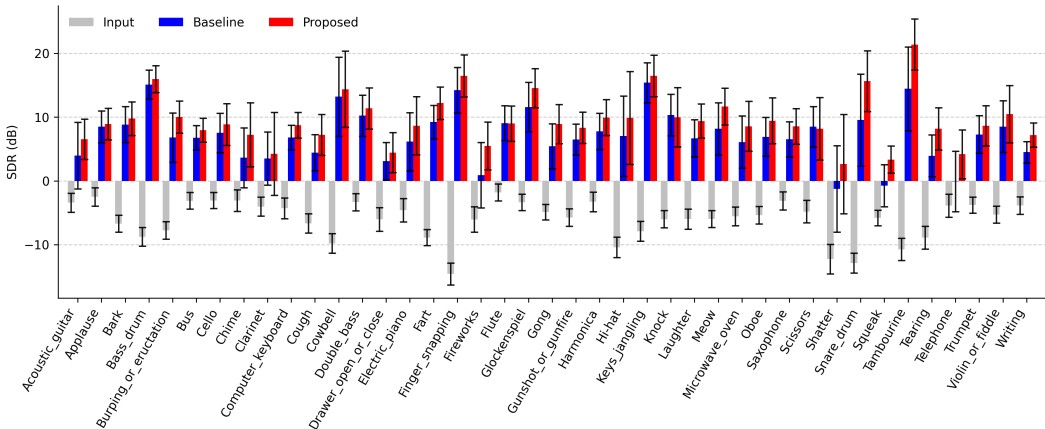

Figure 4: SI-SDR (dB) for each sound event (SE) class, comparing the input mixture, the baseline Sound Selector, and the Sound Selector with proposed method applied. Error bars indicate 95% confidence intervals computed across evaluation samples.

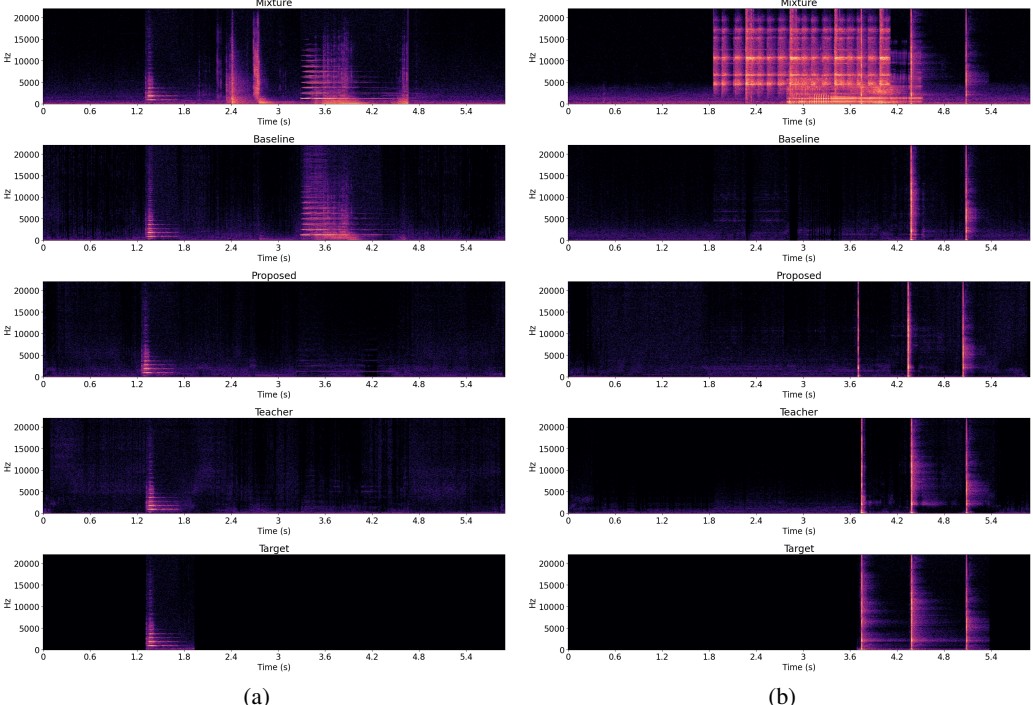

Figure 5: Spectrogram comparisons under a 1-target selection scenario. (a) shows a case where the proposed method achieves improved suppression of interfering sources, while (b) highlights enhanced detection of the target sound

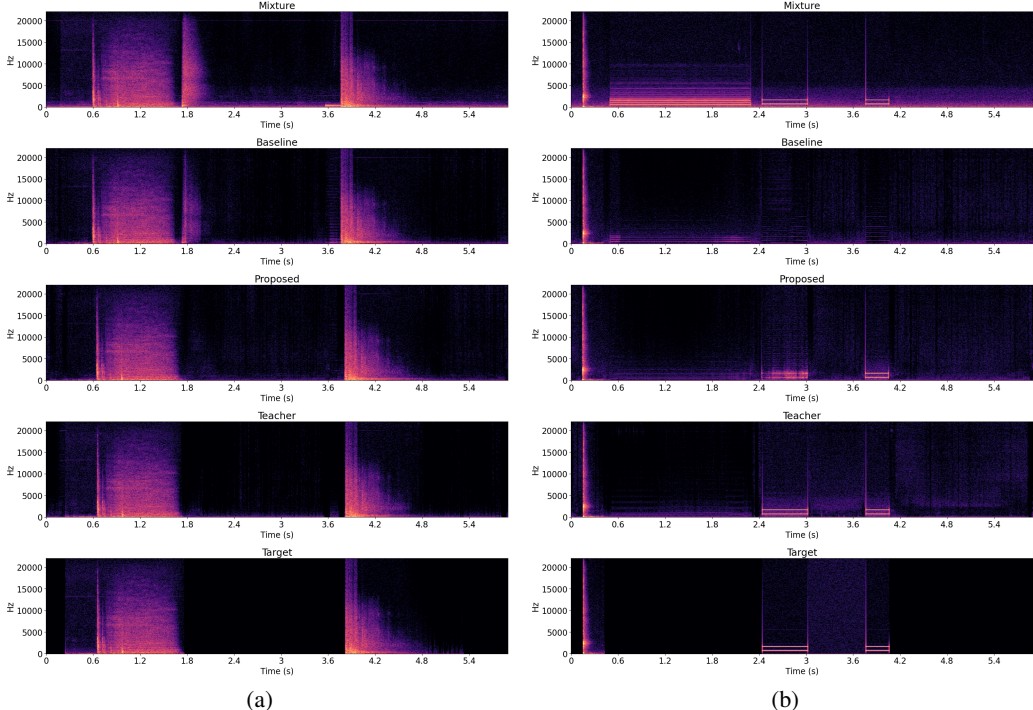

(a)

(b)

Figure 6: Spectrogram comparisons under a 2-target selection scenario. (a) shows a case where the proposed method achieves improved suppression of interfering sources, while (b) highlights enhanced detection of the target sound.

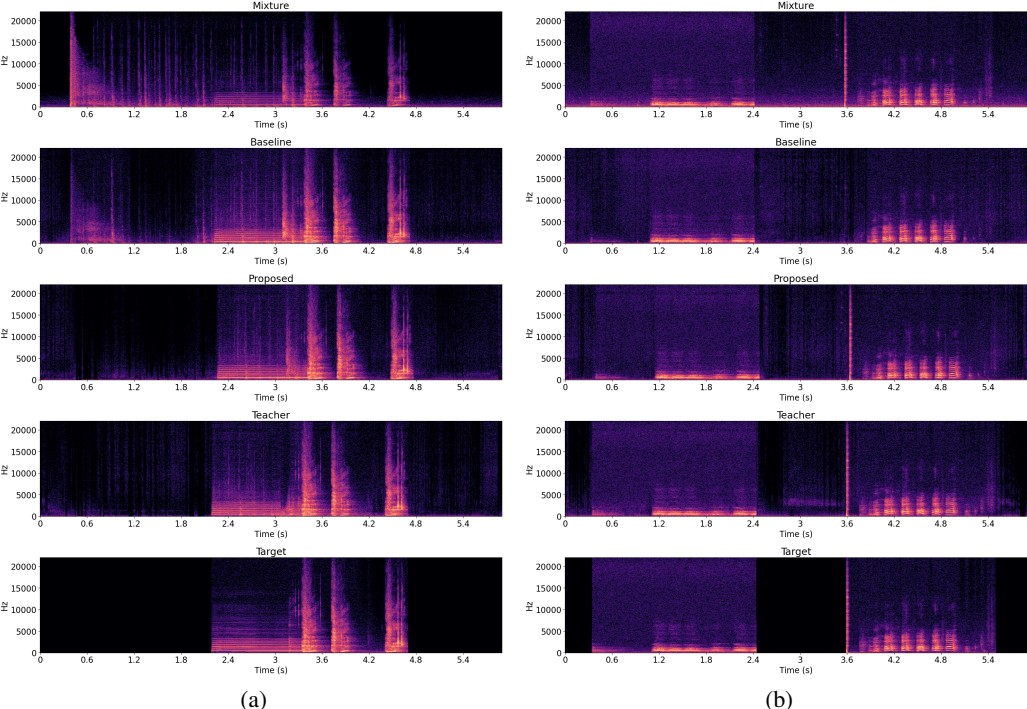

(a)

(b)

Figure 7: Spectrogram comparisons under a 3-target selection scenario. (a) shows a case where the proposed method achieves improved suppression of interfering sources, while (b) highlights enhanced detection of the target sound

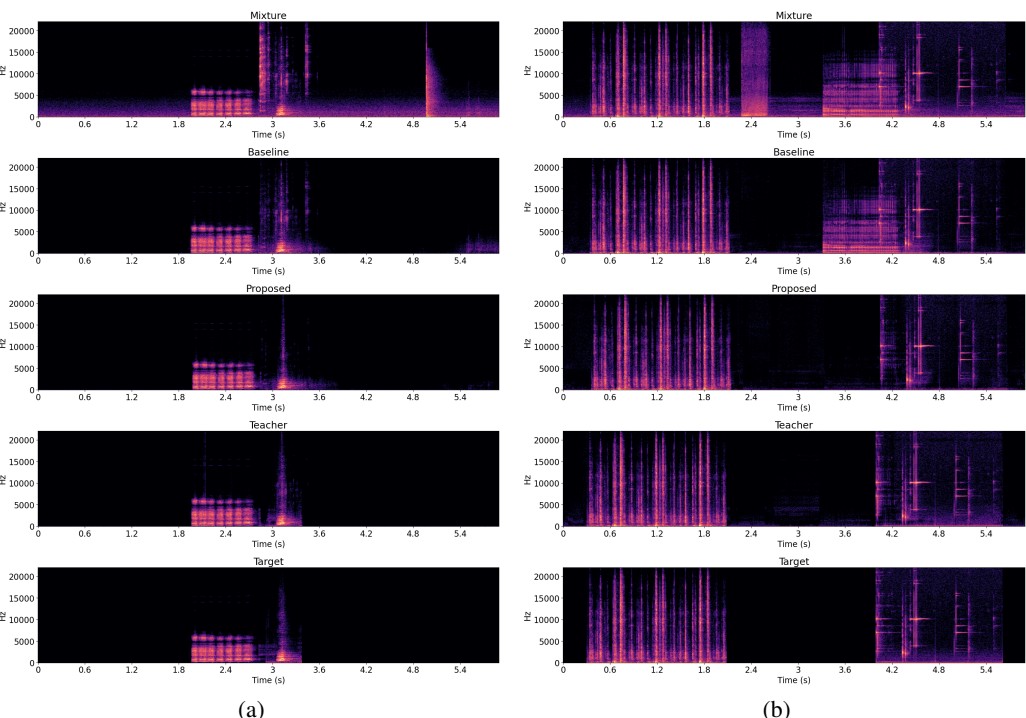

Figure 8: Spectrogram comparisons under target sound overlapping with interfering sources. (a) shows a 2-target selection case, and (b) shows a 3-target selection case. In both scenarios, the proposed method demonstrates improved suppression of overlapping interference compared to the baseline.