# OpenReview forum: "Improving Target Sound Extraction via Disentangled Codec Representations with Privileged Knowledge Distillation"
_NeurIPS.cc/2025/Conference — NeurIPS 2025 poster_

### Official Review · Reviewer_K1JL · 2025-07-03

**Clarity:** 2
**Significance:** 2
**Originality:** 3
**Rating:** 4
**Confidence:** 3

**Summary:**

This paper presents a novel framework, Disentangled Codec Knowledge Distillation (DCKD), to improve Target Sound Extraction (TSE). Recent methods use Privileged Knowledge Distillation (PKD), where a "teacher" model trained with extra information (PI) guides a "student" model. However, existing PKD approaches often suffer from the teacher model overfitting to overly rich PI, leading to ineffective knowledge transfer. DCKD addresses this by regulating the PI's amount and flow. It first uses a neural audio codec to create a compressed representation of the target sound. It then employs Disentangled Representation Learning (DRL) to separate this representation into a static (class-related) factor and a dynamic (class-independent temporal) factor. The teacher model is conditioned in a coarse-to-fine manner: it receives general class information in early layers and the disentangled temporal PI in later layers. This prevents overfitting and facilitates effective feature-level knowledge distillation to the student model. Experimental results show that DCKD consistently improves separation performance across different TSE model architectures, demonstrating that regulating and disentangling privileged information enables more effective knowledge transfer and better student model performance.

**Questions:**

My questions are listed in the weakness part.

**Ethical Concerns:**

["NO or VERY MINOR ethics concerns only"]

**Final Justification:**

Thanks for the response, which resolves:

Clarity issues.
Backbone and baselines selection.
Unresolved concers:

The contribution issue, Since the concept of global and local features, mutual information-based disentanglement, privilege knowledge distillation is widely used in the speech fields, the main contribution of this work are identifying and solving the teacher overfitting issue.
Considering current response, I will adjust my scores to borderline accept.

**Limitations:**

yes

**Paper Formatting Concerns:**

There is no formatting issues.

**Quality:**

3

**Strengths And Weaknesses:**

**[Strengths]**
1. This work proposes a novel method for regulating privileged information using a neural codec for compression and disentangled learning to isolate class-independent temporal details for more effective knowledge transfer, which mitigates the teacher overfitting issue and ensures better student model generalization.
2. The method's effectiveness is empirically validated across multiple distinct model architectures, demonstrating its robustness and general applicability for improving various target sound extraction systems, not just a single setup.

**[Weaknesses]**
1. Clarity. The main motivation of this paper is to mitigating the teacher overfitting issue by disentangling the class-independent temporal details. If my understanding is correct, the overfitting issue of PKD is mainly resulted by the class-dependent informations (the shared features of $l$ and $x$). But currently, the text in Section 1 and 3 does not clearly describe this point, which may raise difficulties in understanding the motivation of this paper quickly. For example, when I read
2. Some out-date modules. Some modules in Figure 2 (a) adopts out-date modules like LSTM and TCN. A minor suggestion is that the authors may try to compare current backbones with Transformer based architecture for the Disentangled Codec and Diffusion Transformer based architecture for the teacher and student model.
3. The baseline selection issue. The authors compare their work with baselines like SoundSelector and WaveFormer in Section 4.3. I agree that this experiment demonstrate the effectiveness of the proposed DCKD method. However, the comparison beyond the proposed framework is missing. For example, the authors could add comparisons with baselines based on other frameworks like SoloAudio [1], AudioSep [2]  (based on Language-oriented TSE), which would make this work more convincing.
4. Since the concept of global and local features, mutual information-based disentanglement, privilege knowledge distillation is widely used in the speech fields, the main contribution of this work are identifying and  solving the teacher overfitting issue.


**Reference:**
[1] SoloAudio: Target Sound Extraction with Language-oriented Audio Diffusion Transformer, Helin Wang et al.
[2] Separate Anything You Describe, Xubo Liu et al.

---

> ### Author Rebuttal · Authors · 2025-07-31
>
> # Response to Reviewer K1JL
>
> We appreciate your thorough and helpful comments. Below are our responses.
>
> **Clarity. The overfitting issue of PKD is mainly resulted by the class-dependent informations (the shared features of  and ). But currently, the text in Section 1 and 3 does not clearly describe this point, which may raise difficulties in understanding the motivation of this paper quickly.**
>
> Thank you for your insightful comment.
>
> What we intended to convey in Sections 1 and 3 is as follows. As mentioned in lines 35–36 of the main paper ("This issue arises because highly predictive PI can lead the teacher to overfit to training signals, producing high-variance outputs that impair the generalization ability of the student."), the overfitting issue in PKD arises when overly rich information is given to the teacher model all at once, without any regulation. In such cases, the teacher can overfit to the privileged information (PI), making it difficult to extract meaningful, generalizable supervision for the student.
>
> In our framework, the overly rich PI corresponds to the target sound itself, which is oracle answer directly. To mitigate this, we compress the target audio using a neural audio codec to retain only the most essential information. Additionally, we disentangle the class information—which is already available in the TSE framework—as a static factor and inject remained dynamic factor into the later part of the teacher model. This way, we separate the information into non-overlapping components (class ID vs temporal), which are delivered progressively and in a regulated manner, reducing the risk of overfitting.
>
> While our disentanglement process involves separate class-dependent information from the codec representations, the core issue is not simply the presence of class-dependent features, but rather the unregulated delivery of overly rich PI to the teacher. The goal of our design is to regularize the teacher’s access to PI in both content and timing.
>
> We appreciate your observation and agree that the current description in Sections 1 and 3 does not fully clarify this point. We will revise the text in the final version to better explain this motivation.
>
> **Some out-date modules. Some modules in Figure 2 (a) adopts out-date modules like LSTM and TCN. A minor suggestion is that the authors may try to compare current backbones with Transformer based architecture for the Disentangled Codec and Diffusion Transformer based architecture for the teacher and student model.**
>
> Thank you for your helpful suggestion.
>
> We used LSTM and TCN modules in our framework because they are well-established and widely adopted in prior work on representation disentanglement (e.g., C-DSVAE [1]) and TSE (e.g., SoundSelector [2]). Our goal was to evaluate the effectiveness of the proposed PKD scheme in a controlled setting.
>
> By demonstrating consistent improvements with conventional backbones, we show that our method is robust and not tied to a specific model architecture. However, we fully agree that transformer-based architectures—such as Diffusion Transformers or TokenMixers—offer great potential.
>
> >[1] J. Bai, W. Wang, and C. P. Gomes, “Contrastively disentangled sequential variational autoencoder,” in Proc. Adv. Neural Inf. Process. Syst., vol. 34, 2021, pp. 10 105–10 118.
>
> >[2] T. Ochiai, M. Delcroix, Y. Koizumi, H. Ito, K. Kinoshita, and S. Araki, “Listen to what you want: Neural network-based universal sound selector,” in Proc. Interspeech, 2020, pp. 1441–1445.
>
> **The comparison beyond the proposed framework is missing. For example, the authors could add comparisons with baselines based on other frameworks like SoloAudio [1], AudioSep [2] (based on Language-oriented TSE), which would make this work more convincing.**
>
> Thank you for the suggestion regarding baseline comparisons.
> Our current baselines (SoundSelector and Waveformer) were selected for their widespread adoption, and compatibility with knowledge distillation. They also represent practical scenarios where lightweight or low-resource deployment is a key consideration.
> We are currently conducting additional experiments applying DCKD to the AudioSep framework. The results will be included in the revised version of the manuscript.
>
> **Since the concept of global and local features, mutual information-based disentanglement, privilege knowledge distillation is widely used in the speech fields, the main contribution of this work are identifying and solving the teacher overfitting issue.**
>
> Thank you for highlighting this point. we will include this point in the revised manuscript.

---

> > ### Author Response · Authors · 2025-08-07
> >
> > Dear Reviewer,
> >
> > Thank you again for your insightful suggestions and comments.
> >
> > As the author-reviewer discussion period finishes on August 8th, we kindly remind you that we won’t be able to follow up after that date. We would appreciate your feedback on whether our responses have addressed your concerns. If you have any further questions or need additional clarification, please let us know before the discussion ends.
> >
> > Thank you for your time.
> >
> >
> > Best regards,
> >
> > The Authors

---

> > ### Comment · Reviewer_K1JL · 2025-08-08
> >
> > Thanks for the response, which resolves:
> > 1) Clarity issues.
> > 2) Backbone and baselines selection.
> >
> > Unresolved concers:
> > 1) The contribution issue,  ``Since the concept of global and local features, mutual information-based disentanglement, privilege knowledge distillation is widely used in the speech fields, the main contribution of this work are identifying and solving the teacher overfitting issue.``
> >
> > Considering current response, I will adjust my scores to borderline accept.

---

> ### Author Response · Authors · 2025-08-08
>
> Thank you for your follow-up comments.
>
> We are glad to know that our responses have addressed most of your concerns.
>
> While concepts such as PKD, global/local features, and mutual information-based disentanglement are indeed widely studied in the speech field, our work applies them to Target Sound Extraction, which involves handling multiple sound classes beyond speech. To the best of our knowledge, this is the first work to introduce PKD together with global/local feature disentanglement and DRL in this domain.
>
> As you noted, addressing the teacher overfitting issue in PKD is also a key contribution of our work. Thank you for highlighting this important point. We will explicitly include it in the main contributions in the revised version.
>
> Thank you again for your time and constructive suggestions, which have helped us further strengthen the paper.

---

### Official Review · Reviewer_dJGw · 2025-07-03

**Clarity:** 3
**Significance:** 3
**Originality:** 3
**Rating:** 5
**Confidence:** 3

**Summary:**

This papers propose a novel privileged knowledge distillation framework for target sound extraction. The authors proposes a coarse-to-fine conditioning mechanism to mitigate overfitting by providing class-level information to earlier layers and class-independent information to later layers. The experimental results demonstrates the effectiveness of the proposed framework in improving the performance of the SoundSelector-based and Waveformer-based student model.

**Questions:**

- (Line 73) "achieving notable gains under acoustically challenging conditions." -> It's unclear what "acoustically challenging conditions" mean here. Please be specific.
- (Figure 2) It would be helpful to highlight which components are being trained and which are frozen.
- (Line 111) $l \in \mathbb{R}^M$ -> Should it be $l \in [0, 1]^M$ to be more precise?
- (Line 146-147) "To further remove class information while preserving fine-grained temporal information, we apply instance normalization," -> It is not straightforward to me how instance normalization can remove class information. Please clarify this.
- (Line 156-157) "To address this, we guide the static factor to retain class-relevant information by minimizing the cosine distance to the embedding vector of the n-hot class label." -> If I understand it correctly, this will make the static factor into class-level features rather than instance-level features, unlike the claimed "temporal-variant" vs "temporal-invariant" features.
- (Table 2) How about the Waveformer architecture? Did you observe a similar trend?
- (Line 331) "Lastly, the removal of feature-level KD loss led to non-trivial performance drops," -> Is this true? The numbers are super close (10.40/9.42 vs 10.40/9.43). Am I missing something here?
- (Line 360-361) "DRL contribute significantly to improving performance over the baseline." -> Please weaken this claim as there's no significance test.

**Ethical Concerns:**

["NO or VERY MINOR ethics concerns only"]

**Final Justification:**

The paper is overall well-written with some minor writing issues. The rebuttal has addressed my main concerns. I am thus recommending accepting this paper.

**Limitations:**

Discussion on the limitations is missing. -> The authors have promised to extend the discussion on the limitations in the revised manuscript.

**Quality:**

3

**Strengths And Weaknesses:**

### Strengths

- The thorough ablation study demonstrates the necessities of each proposed component.
- The experiment results are well-organized and clearly explained.

### Weaknesses

- The proposed method was only evaluated on one dataset and one backbone model for Table 2 and 4 (two backbone models for Table 3). Additional experiments on another dataset would help examine the generalizability of the proposed framework.
- The error bars are missing, making it hard to see if any of these differences are significant.
- The effectiveness of the feature-level KD loss is unclear as the performance remains almost the same when $L_{KD}^{feature}$ is removed.

---

> ### Author Rebuttal · Authors · 2025-07-31
>
> # Response to Reviewer dJGw
>
> Thank you for insightful and constructive reviews. The below is our response:
>
> **Weakness1: The proposed method was only evaluated on one dataset and one backbone model for Table 2 and 4 (two backbone models for Table 3).**
>
> To address the concern, we are currently conducting additional experiments on the ESC-50 dataset [1]. ESC-50 contains 50 sound classes, each with 40 clips (5 seconds each). We split the dataset into 20/10/10 clips per class for training, validation, and test, respectively. Using these, we constructed 1,000 fixed mixtures each for validation and testing. The preliminary results (at epoch 91) show that applying DCKD consistently improves performance over the baseline. These results strengthen the validity of our method. We will include the finalized ESC-50 results and further discussion in the revised manuscript.
>
> |                | SDRi (DB) | SI-SDRi (DB) |
> |:--------------:|:---------:|:------------:|
> |     Student    |    4.55   |     3.47     |
> | Student w DCKD |      4.67     |       3.50       |
>
> > [1] K. J. Piczak, “ESC: dataset for environmental sound classification,” in Proc. ACM Multimed., 2015.
>
> **Weakness2: The error bars are missing, making it hard to see if any of these differences are significant.**
>
> We appreciate the suggestion. We will revise the figures to include error bars for all quantitative results to improve clarity.
>
> **Weakness3: The effectiveness of the feature-level KD loss is unclear as the performance remains almost the same when L_kd^feature is removed.**
>
> We will revise the text, weakening the original claims.
>
> Although the improvement appears modest, the feature-level KD loss serves as a meaningful regularizer during training. Specifically, it encourages alignment between the intermediate representations of the student and the teacher, which helps stabilize learning dynamics and facilitates faster convergence. For example, we observed that the model with feature-level loss converged at epoch 287, compared to epoch 296 without it.
> We acknowledge that the improvement in final metrics is not substantial enough to justify claims. We will revise the text accordingly.
>
> **Questions:**
>
> **(Line 73) "achieving notable gains under acoustically challenging conditions." -> It's unclear what "acoustically challenging conditions" mean here. Please be specific.**
>
> Thank you for pointing this out. We will revise the sentence for clarity:
> → “achieving notable gains under acoustically challenging conditions, such as mixtures containing 0 to 3 target sounds with varying background noise.”
>
> **(Figure 2) It would be helpful to highlight which components are being trained and which are frozen.**
>
> We are very thank you for pointing this out. We will revise Figure 2 to visually indicate frozen and trainable components. Specifically, modules shaded in blue are frozen during training.
>
> **(Line 111)**
>
> Thank you for the suggestion. we will revise this one on the final version.
>
> **(Line 146-147) "To further remove class information while preserving fine-grained temporal information, we apply instance normalization," -> It is not straightforward to me how instance normalization can remove class information. Please clarify this.**
>
> Thank you for the detailed review. The use of instance normalization (IN) in our framework aims to reduce class-specific bias by normalizing the first- and second-order statistics (mean and variance) of each instance. This technique is widely used in the style transfer literature[2], where it has been shown to effectively remove speaker ID information by suppressing global feature trends.
> In the context of sound representation, we assume that class-related cues are embedded in global activation patterns (e.g., high energy regions specific to certain classes). IN mitigates such biases by enforcing zero-mean and unit-variance across feature maps, thereby encouraging the model to retain only temporally localized, class-agnostic details. This statistical filtering complements our overall disentanglement process in the DSDE module. We will revise the text to make this intuition more explicit.
>
> We will clarify this in the revised manuscript.
>
> **(Line 156-157) "To address this, we guide the static factor to retain class-relevant information by minimizing the cosine distance to the embedding vector of the n-hot class label." -> If I understand it correctly, this will make the static factor into class-level features rather than instance-level features, unlike the claimed "temporal-variant" vs "temporal-invariant" features.**
>
> The static factor is explicitly encouraged to encode class-level, temporally invariant characteristics by minimizing cosine distance to a class embedding vector. This aligns with our design:
> - The static factor captures global, class-consistent features.
> - The dynamic factor models temporal, instance-specific variations.
>
> **(Table 2) How about the Waveformer architecture? Did you observe a similar trend?**
>
> Thank you for your question. Due to computational resource constraints, we were unable to conduct a full ablation study on the Waveformer architecture within the scope of the main paper.
> However, we agree that analyzing whether the same trend holds for Waveformer is important. We are currently conducting additional experiments and will include the results of ablation study for the Waveformer in the supplementary material.
>
> **(Line 360-361) "DRL contribute significantly to improving performance over the baseline." -> Please weaken this claim as there's no significance test**
>
> Thank you for pointing this out. We will revise the manuscript to weaken the phrasing—e.g., replacing “contribute significantly” with “consistently improve”—to more accurately reflect the experimental evidence.
>
> **Discussion on the limitations is missing.**
>
> We included our limitations in the Conclusion section.

---

> ### Comment · Reviewer_dJGw · 2025-08-05
>
> Thank you for the detailed rebuttal.
>
> - **Weakness 1**: Thanks for providing additional experimental results. These would definitely help strengthen the claims.
> - **Weakness 3**: Sounds good
> - **Line 146-147**: Thanks for the clarification. Why do you think the assumption that "class-related cues are embedded in global activation patterns (e.g., high energy regions specific to certain classes)" is always true? I don't think this is a verified argument, and the current writing is somewhat misleading. Please rephrase the writing and state explicitly the assumptions here.
> - **Line 156-157**: "Class-level vs. instance-level" and "temporal-variant vs. temporal-invariant" are two separate concepts. Some instance-level features can still be temporal-invariant, and thus disentangling "class-level and instance-level" features is different from disentangling "temporal-variant and temporal-invariant" features. This sentence is not wrong, but the two concepts have been used interchangeably in several places.
> - **Limitations**: More discussions on the limitations are needed.

---

> > ### Author Response · Authors · 2025-08-06
> >
> > Thank you for your follow-up comments.
> >
> > **(Line 146-147)**
> >
> > We acknowledge that the assumption regarding class-related cues being embedded in global activation patterns may be misleading.
> >
> > As noted previously, Instance Normalization (IN) is widely used in style transfer and voice conversion literature to filter out global style attributes (such as illumination and contrast in images [1], or pitch and energy in audio [2]) while preserving content information.
> >
> > Based on these findings, we hypothesize that IN in our framework can filter out class-specific global features, such as characteristic pitch or energy associated with certain sound classes.
> >
> > Since our ablation studies show that using IN improves performance, we interpret this as evidence that IN facilitates the disentanglement of class-related information from the codec representation, thereby enabling DCKD to be applied more effectively. We will revise the manuscript to explicitly present this as a hypothesis, supported by both prior work and our empirical findings.
> >
> > > [1] X. Jin, C. Lan, W. Zeng, Z. Chen, and L. Zhang, “Style Normalization and Restitution for Generalizable Person Re-Identification,” presented at the Proceedings of the IEEE/CVF Conference on Computer Vision and Pattern Recognition, 2020, pp. 3143–3152.
> >
> >
> > > [2] W. Quamer and R. Gutierrez-Osuna, "End-To-End Streaming Model For Low-Latency Speech Anonymization," 2024 IEEE Spoken Language Technology Workshop (SLT), Macao, 2024, pp. 727-734
> >
> >
> > **(Line 156-157)**
> >
> > As you pointed out, we acknowledge that our previous explanation may have caused some confusion regarding the distinction between temporal-level and instance-level representations.
> >
> > First, we would like to clarify that the term “instance-level” does not appear anywhere in our manuscript. We assume the reviewer’s concern has arisen from our use of instance normalization (IN), which might have been interpreted as a mechanism for extracting temporal information.
> >
> > We would like to clarify the intended role of instance normalization in our framework.
> > IN is not specifically designed to extract temporal information. Instead, it is applied to filter out class-related information from the codec representation as an initial step toward disentanglement.
> >
> > The output of the dynamic encoder is interpreted as containing temporal information because it captures the residual features after class-related components have been removed from the codec representation, which includes both class-related and temporal information.
> > Moreover, the dynamic encoder is implemented with an LSTM architecture, which is inherently capable of modeling temporal dynamics. This architectural choice further supports the interpretation of its output as encoding temporally varying features.
> >
> > We thank the reviewer for raising this point and will revise the manuscript accordingly to clarify this distinction.
> >
> > **Limitations**
> >
> > Regarding the limitations, we plan to describe them more in the revised manuscript. Specifically, as mentioned in the rebuttal, our current method assumes a closed-set setting where a reference target sound is available for each mixture, which may limit the usage of real-world mixtures during training. Additionally, we will elaborate in the appendix on the increased training time and parameter overhead introduced by integrating DCKD into existing architectures.

---

> > > ### Comment · Reviewer_dJGw · 2025-08-06
> > >
> > > - **Line 146-147**: Thanks for the clarification.
> > > - **Line 156-157**: Sorry for the confusion. I am referring to "class-independent temporal information", which appears many times in the manuscript. My point is that "class-independent" and "temporal" are two separate concepts. The proposed loss function only encourages the disentanglement of "class-variant vs class-independent" features but *not* "temporal-variant vs. temporal-invariant" features. It's misleading to put these words together as if they are always the same concept without explicitly stating the underlying assumptions. Anyway, I think some clarification in the revised manuscript would help clarify this.
> > > - **Limitations**: Sounds good. That would strengthen the paper.
> > >
> > > I think all my main concerns have now been addressed, and I'll raise my rating to reflect that! Thank you for all the clarifications and follow-ups.

---

> > > > ### Author Response · Authors · 2025-08-07
> > > >
> > > > Thank you for your follow-up comments and clarification.
> > > >
> > > > We define the static and dynamic factors in our framework by adapting the concepts described in [1] to the sound domain. In particular, [1] explains:
> > > > > “representations of a video recording the movements of a cartoon character could be disentangled into the character identity (static) and the actions (dynamic). For audio data, the representations shall be able to separate the speaker information (static) from the linguistic information (dynamic). ” [1]
> > > >
> > > > Based on this description, we interpret the static factor as class information and the dynamic factor as temporal information, and apply disentangled representation learning accordingly.
> > > >
> > > > However, we acknowledge that in our manuscript, the term “class-independent temporal information” was used without sufficient explanation, which may cause confusion. We will revise the manuscript to explicitly clarify this point.
> > > >
> > > > Thanks again for acknowledging our response, and we are glad to know that your concerns have been addressed.
> > > >
> > > > - Reference:  [1] J. Bai, W. Wang, and C. P. Gomes, “Contrastively Disentangled Sequential Variational Autoencoder,” in Advances in Neural Information Processing Systems, Curran Associates, Inc., 2021, pp. 10105–10118.

---

### Official Review · Reviewer_TrbU · 2025-07-03

**Clarity:** 3
**Significance:** 3
**Originality:** 2
**Rating:** 4
**Confidence:** 4

**Summary:**

The paper proposes a method for Target Sound Extraction in a PKD frameword that overcomes the possibility of overfitting that usually happens in PKD. The key contributions are generating a disentangled representation of the target sound. The disentanglement also acts as a regularization that helps avoid overfitting. In order to learn meaningful disentanglement, mutual information objectives are used.

**Questions:**

1. What makes this problem definition only limited to the scope of TSE? Is the solution generalizable to any domain or are we making certain assumptions to make the solution work only for TSE.
2. What is the effect of the codec model on the performance? It would be nice to analyze a variety of neural codec models and understand its impact.

**Ethical Concerns:**

["NO or VERY MINOR ethics concerns only"]

**Final Justification:**

I would like to stand by my original decision of borderline accept.  My primary concerns with the paper were the limited datasets and the lack of comparison to other baselines. The authors in the rebuttal mentioned that more datasets will be added in the revised paper. Although they mentioned that additional baselines is not feasible, this paper has enough contributions to justify a borderline accept.

**Limitations:**

I was not able to find a limitations section in the paper. In the checklist, it is mentioned that section 6 has the limitations, but I am not able to find a section 6.

**Paper Formatting Concerns:**

I dont see any major formatting concerns.

**Quality:**

3

**Strengths And Weaknesses:**

Strengths:
1. The problem is well-motivated and the proposed solution is explained clearly. Overall the paper is well-written and easy to understand.
2. The authors provide ablation studies explaining the effect of each of the proposed component.
3. The problem the paper is trying to solve is significant: reducing overfitting in PKD
4. Although the proposed methodologies are not novel, it is a good solution to the problem in hand.

Weakness:
1. The authors do not compare the proposed method with other TSE models/baselines from related work. From my understanding, table 3 only compares distillation based methods w/wo the proposed solution.
2. The experimental section is also limited to one synthetic dataset that the authors curate. To understand the true potential of this model, we need to experiment with more datasets.

---

> ### Author Rebuttal · Authors · 2025-07-31
>
> # Response to Reviewer TrbU
>
> Thank you for your insightful feedbacks. Below, we provide responses to each of your points.
>
> **Weakness1: table 3 only compares distillation based methods w/wo the proposed solution.**
>
> Thank you for your valuable feedback. We understand the importance of situating our proposed method in the broader context of prior work.
>
> To clarify:
> The “student” models presented in Table 3 are based on existing TSE architectures—SoundSelector and Waveformer—which were proposed in prior works. The “student” entries correspond to the original TSE baselines without any knowledge distillation or privileged information, while the “w/ DCKD” entries reflect performance after applying our proposed method to those existing models.
>
> Thus, Table 3 already shows that our method provides consistent improvements over established, non-distilled TSE baselines across model types and mixture conditions. We chose these particular models as baselines because they are representative, widely-used, and lightweight enough to be distilled effectively.
> We will revise the Table 3 to include citations for these models.
>
> **Weakness2: The experimental section is also limited to one synthetic dataset that the authors curate.**
>
> To address the concern, we are currently conducting additional experiments on the ESC-50 dataset [1]. ESC-50 contains 50 sound classes, each with 40 clips (5 seconds each). We split the dataset into 20/10/10 clips per class for training, validation, and test, respectively. Using these, we constructed 1,000 fixed mixtures each for validation and testing. The preliminary results (at epoch 91) show that applying DCKD consistently improves performance over the baseline. These results strengthen the validity of our method. We will include the finalized ESC-50 results and further discussion in the revised manuscript.
>
> |                | SDRi (DB) | SI-SDRi (DB) |
> |:--------------:|:---------:|:------------:|
> |     Student    |    4.55   |     3.47     |
> | Student w DCKD |      4.67     |       3.50       |
>
> > [1] K. J. Piczak, “ESC: dataset for environmental sound classification,” in Proc. ACM Multimed., 2015.
>
> **Q1. What makes this problem definition only limited to the scope of TSE? Is the solution generalizable to any domain or are we making certain assumptions to make the solution work only for TSE.**
>
> Thank you for your insightful question. While our proposed method was developed and evaluated in the context of Target Sound Extraction (TSE), we believe the core idea behind our method is not inherently limited to this domain.
>
> Disentangled Codec-based Knowledge Distillation (DCKD) is motivated by a key insight:
> When applying Privileged Knowledge Distillation (PKD), providing overly rich or entangled privileged information (PI) all at once can lead to suboptimal student learning. Instead, regulating the flow of PI by disentangling it into static (e.g., class-info) and dynamic (e.g., temporal info) factors, and injecting them at early and later of the teacher model, allows for more effective supervision.
>
> We believe this strategy is not specific to sound classes—it reflects a general principle: progressive and structured use of privileged information improves distillation.
> For instance:
>
> - In *target speaker extraction*, a target speaker’s utterance can be used as PI, disentangled into speaker identity and temporal dynamics.
>
> - In *audio-visual separation*, video frames could provide PI with spatial-static and motion-dynamic cues.
>
> - In *cross-modal generation or retrieval*, language descriptions could be disentangled into semantics, syntax, or style components.
>
> We will elaborate on this generalization potential in the revised version. Thank you again for pointing out this important aspect.
>
> **Q2. What is the effect of the codec model on the performance? It would be nice to analyze a variety of neural codec models and understand its impact.**
>
> We appreciate the suggestion to analyze the influence of different codec models. In our current work, we adopted the DAC codec primarily due to its strong reconstruction quality and publicly available pretrained model, which allowed us to focus on evaluating the DCKD framework itself.
>
> We fully agree that the choice of codec may significantly affect the type and fidelity of information passed as privileged input to the teacher model. For instance, lower-bitrate codecs may yield more compressed but semantically informative features, while high-bitrate codecs may have fine-grained details. Understanding the effect on DCKD performance of this trade-off would be helpful.
>
> Due to limited resources, we were not able to conduct a comprehensive study across multiple codecs within this submission. However, we recognize this as a key direction for future work.
>
> **Limitations: I was not able to find a limitations section in the paper. In the checklist, it is mentioned that section 6 has the limitations, but I am not able to find a section 6.**
>
> Thank you for pointing this typo. Limitation is included in Section 5 (Conclusion).

---

> > ### Author Response · Authors · 2025-08-07
> >
> > Dear Reviewer,
> >
> > Thank you again for your insightful suggestions and comments.
> >
> > As the author-reviewer discussion period finishes on August 8th, we kindly remind you that we won’t be able to follow up after that date. We would appreciate your feedback on whether our responses have addressed your concerns. If you have any further questions or need additional clarification, please let us know before the discussion ends.
> >
> > Thank you for your time.
> >
> >
> > Best regards,
> >
> > The Authors

---

> > ### Comment · Reviewer_TrbU · 2025-08-08
> >
> > Thanks for the response. I have no further questions.

---

### Official Review · Reviewer_Vu2K · 2025-07-03

**Clarity:** 3
**Significance:** 4
**Originality:** 4
**Rating:** 5
**Confidence:** 5

**Summary:**

This paper introduces a novel Disentangled Codec Knowledge Distillation (DCKD) framework for target sound extraction (TSE). TSE is the task of isolating a specific sound source from a mixture given a clue about the target (e.g., class label or reference). The authors leverage privileged knowledge distillation (PKD) – a strategy where a teacher model has access to additional information (privileged information, PI) during training that the student model doesn’t get at test time. The key idea in DCKD is to carefully regulate the amount and flow of target information provided to the teacher: they compress the target audio through a neural audio codec to limit information, and apply disentangled representation learning (DRL) to separate class-related vs. fine-grained details. The teacher model is trained with both the usual target class label (an n-hot vector indicating which sound classes to extract) and the disentangled codec representation of the target audio as PI. The student model, which at test time only receives the class label, learns from the teacher via feature-level knowledge distillation. Experiments on a multi-target sound extraction dataset (synthetic mixtures from FSD Kaggle2018 + TAU Urban Acoustic Scenes 2019) show that DCKD yields consistent improvement in signal-to-distortion ratio (SDR) for the student model across different architectures and difficult scenarios. Notably, DCKD outperforms prior PKD approaches that use richer PI (like full target audio) by avoiding teacher overfitting and achieving better student generalization.

In summary, the DCKD approach integrates these pieces: privileged info in the form of a compressed, disentangled target audio representation, delivered to the teacher in a coarse-to-fine manner and then apply feature level knowledge distillation to train the student.

**Questions:**

The paper should better explain how the privileged information is integrated into the teacher model. Currently, it’s only stated that the n-hot label conditions early layers and the disentangled dynamic factors condition later layers. However, it’s unclear how this conditioning is implemented. The architecture details should be cleared in the supplementary material for the paper if not in the main paper to share the details.

The authors themselves note that the use of DAC introduces a significant computational increase. It would be great if authors can detail the overhead from model size, training time details and discuss the computational requirements.  For instance, how much longer (in epochs or wall-clock time) does training DCKD take compared to a baseline? Was the teacher model significantly larger in parameters or just in input dimension?

The proposed PKD framework assumes access to isolated target audio examples for every sound class during training. In the synthetic experiment this is feasible (mixtures are generated with known sources), but in real-world scenarios it might be hard to obtain clean target samples for every sound of interest. For example, is it assumed that one has a collection of prototypical sounds for each class (as in FSD Kaggle) to use as PI? If so, the authors should clarify this and perhaps note that the method is limited to closed-set target classes. Relatedly, how well would the student model generalize to a new sound class that was not seen in training or had no privileged example? Since the student only uses class labels, in principle it could handle new classes if trained appropriately – but without a teacher for those, it’s unclear.

**Ethical Concerns:**

["NO or VERY MINOR ethics concerns only"]

**Final Justification:**

The authors' rebuttal and follow-up have thoroughly addressed my concerns regarding PI integration, computational details, and assumptions/limitations on isolated target audio and generalization. The clarifications on training overhead and future directions (e.g., lightweight compression, weak supervision, cross-domain applications) mitigate the weaknesses I noted, while reinforcing the strengths of the DCKD framework's novelty, validation across architectures, and practical inference efficiency.  Additionally, the authors' commitments to including new experiments on ESC-50, further baselines like AudioSep, and expanded discussions on limitations (e.g., closed-set assumptions, real-world applicability) address points raised by other reviewers, contributing to a more robust revised manuscript.

This is a technically solid paper with good impact in audio processing sub-areas, good evaluation, and no ethical issues. I stand by my original accept recommendation.

**Limitations:**

The paper itself notes one main limitation: the reliance on the heavy DAC (neural codec) and the additional training cost. A future direction they plan is to design a lightweight feature compression method to serve the role of DAC. Perhaps a smaller autoencoder trained jointly could replace the off-the-shelf codec, or dimensionality reduction techniques could be used to create a bottleneck without a full codec.

It would also be interesting to apply DCKD to speech domain explicitly (target speaker extraction). In that case, the privileged info could be a clean recording of the target speaker’s voice. Actually, that scenario is common (speaker extraction often assumes a reference voice recording is available at runtime). But if we assume a case where at runtime you only know the speaker’s identity or name (not a voice snippet), one could train a teacher with a reference voice audio (privileged) and then distill a student that only needs speaker ID (like a class label). This mirrors what DCKD does (because in DCKD student only gets class label, teacher heard the actual voice). This could allow speaker extraction without needing enrollment audio at test time, which is a powerful capability. The technique would be largely the same (codec + DSDE on the voice). This cross-application potential is worth exploration.

Real world application: The dataset here is synthetic. Real recordings might have more variation (reverberation, non-additive noise, etc.). The method might need to prove itself on real sound separation benchmarks (though few exist for arbitrary classes). Possibly, using datasets like AVE (Audio-Visual Event) or overlapping ESC-50 might be a next step. The reliance on a known set of classes and available example recordings per class is a constraint – expanding beyond class-conditioned separation to, say, text-based queries (like “extract the sound of glass breaking”) could be a future extension. Perhaps a language model could generate an “imagined” audio representation as privileged info for a teacher, but that’s quite speculative.

**Paper Formatting Concerns:**

The submission includes the required checklist and follows the general format. I didn’t spot formatting issues; tables and figures were clear and referenced.

**Quality:**

3

**Strengths And Weaknesses:**

Strengths:

This work extends PKD to a new domain (general sound separation) with a much richer form of PI (actual audio content) than prior TSE studies. The coarse-to-fine conditioning strategy is insightful it acknowledges that not all information should be dumped in at once. By structuring the flow (class clue first, then details), the teacher’s utilization of privileged data is more controlled. This is a non-trivial improvement over earlier PKD approaches that might simply concatenate an extra feature and risk overfitting.

The authors validate DCKD on two model architectures and different numbers of target sources. The consistent gains (approximately +1 to +2 dB SI-SDRi across the board) show the method is not tied to a specific network or scenario. This implies the approach will be robust under a range of scenarios. Moreover, demonstrating improvement even for a lightweight real-time model (Waveformer) is important: it indicates DCKD can help in practical settings where computational resources are limited.

The inference overhead is also low since student model has the same input requirements and the model size is same as the TSE model where the approach improves the performance without increasing the runtime cost or the model size for the deployment.

Weakness:

Computational inefficiency: The DCKD framework is quite complex. It requires training a pretrained neural codec, a DSDE module with multiple loss terms and and a large teacher model, followed by a student. This is overall computationally heavy in case the models are not available. The teacher training runs for up to 500 epochs. In comparison, simpler PKD methods (like using just a timestamp) are much easier to implement and train. The authors acknowledge the computational inefficiency as a limitation and propose to work on a lightweight compression model in future.

The privileged data used here is specifically audio of the target source itself. This is a strong assumption – essentially, one needs isolated target recordings for training as PI (which they simulate by knowing the mix composition). In real applications, obtaining a clean example of the exact sound one wants to extract (for training) may not be straightforward. By contrast, using timestamps or class names as PI (like prior works) doesn’t require providing the sound itself. The paper demonstrates the superiority of using the target audio as PI (via codec) over those simpler PI types but it doesn’t solve how to get such audio in a realistic setting unless synthetic data or extra sensors are available. If one has a library of isolated sounds for each class, one could use that, but then generalizing to new sounds not in the library might be an issue. In real life, training with real mixtures might not come with ground-truth isolated tracks unless it’s a task like speech where data can be simulated. This is not a flaw in the paper but rather a technical limitation on the application of this method. Would be great to see author's comments on this aspect.

Clarification needed: The authors did not report standard deviations or statistical significance. Given that they use a fixed test set of 5000 mixtures, this is fine, but some indication of variance (perhaps via multiple runs) could strengthen confidence. In the NeurIPS checklist, they marked that they reported statistical significance, but in the paper’s content, we mainly see mean SDRi. However, since improvements are fairly large relative to any plausible variance, this is a minor issue.

---

> ### Author Rebuttal · Authors · 2025-07-31
>
> # Response to Reviewer Vu2K
>
>
> We are very grateful for your valuable comments and suggestions. Below is our response to your review:
>
> **Q1. The paper should better explain how the privileged information is integrated into the teacher model.**
>
> Thank you for the comment.
> The n-hot label is injected into the right after the learnable encoder of the model via element-wise multiplication, where the label vector is first linearly projected to match the channel dimension of the latent feature. The dynamic factors are injected into the latter part through element-wise multiplication, specifically right before the output layer, which consists of a conv1d layer. Thank you again for highlighting this important clarification. We will clarify this in the revised manuscript.
>
> **Q2. how much longer (in epochs or wall-clock time) does training DCKD take compared to a baseline? Was the teacher model significantly larger in parameters or just in input dimension?**
>
> Thank you for the suggestion. We will elaborate on this part in the revised manuscript.
>
> 1. Model Size: As we reported in Table 3, the parameter size of teacher model is increased due to DAC parameter size, about 80M.
> 2. Training Time:
> Wall-clock time increases when training a student with DCKD by approximately 2.7 times compared to training a baseline student model alone, primarily due to the additional forward passes through the teacher model and the computation of the mutual information-based loss terms.
>
>     - Teacher model: 3 days
>     - Student model: 9 hours
>     - Student with DCKD: 1 day
>
>       on 4 NVIDIA GeForce RTX 4090 GPUs.
>
> **Q3. Is it assumed that one has a collection of prototypical sounds for each class (as in FSD Kaggle) to use as PI? If so, the authors should clarify this and perhaps note that the method is limited to closed-set target classes. Relatedly, how well would the student model generalize to a new sound class that was not seen in training or had no privileged example? Since the student only uses class labels, in principle it could handle new classes if trained appropriately – but without a teacher for those, it’s unclear.**
>
>  *About the reference audio during training*
>
> Thank you for your insightful comment. Yes. Our framework assumes access to reference audio for each mixture during training. This assumption aligns with the common practice in the target sound extraction (TSE) domain, where supervised training typically requires isolated target signals as ground truth. As you mentioned,  this is a limitation of the general supervised learning paradigm for sound extraction, rather than being specific to our method.
>
> We agree that performing training with real mixtures without access to target recordings is practically important challenge. In such cases, we believe a feasible solution would be to pretrain the model using a synthetic dataset that includes the target sound class and is constructed to resemble real-world acoustic conditions as closely as possible. The pretrained model can then be deployed for inference on real mixtures.
>
> Furthermore, [1] has demonstrated that sound separation can be learned under weak supervision using only frame-level or clip-level labels. This type of weak PI could be incorporated into our framework in future work.
> Moreover, your suggestion to explore other forms of PI, such as text-based descriptions or language model-derived embeddings, is also inspiring.
>
> > [1] F. Pishdadian, G. Wichern, and J. Le Roux, “Finding Strength in Weakness: Learning to Separate Sounds With Weak Supervision,” IEEE/ACM Transactions on Audio, Speech, and Language Processing, vol. 28, pp. 2386–2399, 2020.
>
> *Regarding generalization*
>
> As the current model is conditioned on a fixed set of 41 predefined classes, the input channel size of the class embedding layer is correspondingly fixed. Therefore, to generalize to new classes not seen during training, the model architecture needs to be modified and fine-tuned accordingly. This is an inherent limitation of class-conditioned models. Exploring more flexible conditioning mechanisms—such as continuous or text-based queries—that could allow generalization beyond fixed class sets would be interesting future work.
>
>
> **Future directions: lightweight feature compression instead of DAC, domain trasformation (target speaker extraction), Real world application, text-based queries**
>
> Thank you for the suggestions. We are actively exploring the directions you mentioned.

---

> > ### Author Response · Authors · 2025-08-07
> >
> > Dear Reviewer,
> >
> > Thank you again for your insightful suggestions and positive comments.
> >
> > As the author-reviewer discussion period finishes on August 8th, we kindly remind you that we won’t be able to follow up after that date. We would appreciate your feedback on whether our responses have addressed your concerns. If you have any further questions or need additional clarification, please let us know before the discussion ends.
> >
> > Thank you for your time.
> >
> >
> > Best regards,
> >
> > The Authors

---

### Decision · Program_Chairs · 2025-09-17

**Decision:**

Accept (poster)

**Comment:**

Summary:

The paper proposes a method for target sound extraction in a privileged knowledge distillation (PKD) framework. The main idea is to regulate the amount of target information provided to the teacher through a compressed representation of the target sound and static/dynamic (or coarse/fine) information disentanglement. The experimental results are convincing and there are a number of insightful ablations, mentioned by the reviewers.

Strengths:

- Well-motivated topic, well-written paper, well-organized results.
- Extension of PKD to example-based general sound separation.
- Consistent gains across the board.
- Nice ablations.

Weaknesses:

- Inefficient framework.
- Unrealistic assumption about having a totally clean audio as query, thus "forcing" training/results on synthetic data (but this is more of a task setup than a problem of the specific approach).
- Potential lack of generality, which the authors address with an additional experiment on ESC-50.

Reasons for accept/reject:

All reviewers agree this submission is of enough quality and that the approach could be interesting/relevant to researchers in the field. The authors also highlight the potential application of the developed concepts to related fields such as target speaker extraction, audio-visual separation, and cross-modal generation/retrieval.

Discussions:

The authors did a good rebuttal, although they already had received quite good scores at the beginning. I'd urge the authors to include the additional ESC-50 results in the main paper.